# Quantifying risk factors associated with light-induced potato tuber greening in retail stores

**Sabine Tanios**[1,2], **Alieta Eyles**[1,2], **Ross Corkrey**[3], **Robert S. Tegg**[2],
**Tamilarasan Thangavel**[2], **Calum R. Wilson**[1,2]*

**1** ARC Training Centre for Innovative Horticultural Products, Tasmanian Institute of Agriculture, University of Tasmania, New Town Research Laboratories, New Town, Tasmania, Australia, **2** Tasmanian Institute of Agriculture, University of Tasmania, New Town Research Laboratories, New Town, Tasmania, Australia, **3** Tasmanian Institute of Agriculture, University of Tasmania, Hobart, Tasmania, Australia

* calum.wilson@utas.edu.au

**Data Availability Statement:** All relevant data are within the paper and its Supporting Information files.

## Abstract

Light conditions in retail stores may contribute to potato greening. In this study, we aimed to develop a potato tuber greening risk rating model for retail stores based on light quality and intensity parameters. This was achieved by firstly exposing three potato varieties (Nicola, Maranca and Kennebec) to seven specific light wavelengths (370, 420, 450, 530, 630, 660 and 735 nm) to determine the tuber greening propensity. Detailed light quality and intensity measurements from 25 retail stores were then combined with the greening propensity data to develop a tuber greening risk rating model. Our study showed that maximum greening occurred under blue light (450 nm), while 53%, 65% and 75% less occurred under green (530 nm), red (660 nm) and orange (630 nm) light, respectively. Greening risk, which varied between stores, was found to be related to light intensity level, and partially explained potato stock loss in stores. Our results from this study suggested that other in-store management practices, including lighting duration, average potato turnover, and light protection during non-retail periods, likely influence tuber greening risk.

## Introduction

Following light exposure, potato tubers turn green, due to the accumulation of chlorophyll below the skin. Tuber greening is a major cause of quality loss, which can occur at any stage along the supply chain, and therefore managing losses due to greening is a significant challenge for the potato industries [1–2]. Consumer rejection of green potatoes can be attributed to a perceived link between greening and the presence of glycoalkaloids, which occurs via a light-induced metabolic pathway independent to that responsible for chlorophyll production [2]. Protecting tubers from light exposure, especially in retail stores, is problematic as consumers prefer unpackaged potatoes and/or clear packaging [3], and as such, higher greening incidence could occur.

Light duration and intensity have been shown to affect the rate of greening [4–5]. Greening is an accumulative process, whereby the longer tubers are exposed to light, the greater greening occurs. In general, increases in light intensity have been shown to increase greening rates [6–

**Funding:** CRW and ST received funding from the Australian Research Council's Industrial Transformation Training Centres scheme under Grant IC140100024 (htpps://arc.gov.au). The funder had no role in study design, data collection and analysis, decision to publish, or preparation of the manuscript.

**Competing interests:** The authors have declared that no competing interests exist.

8]. A light intensity of 750 lx (10 μmol m$^{-2}$ s$^{-1}$) was sufficient to induce maximum greening, while exposure above this value, 750–1250 lx (10–17 μmol m$^{-2}$ s$^{-1}$), did not increase greening [4].

Lighting types and spectral composition have also been shown to influence greening. For example, the rate of chlorophyll accumulation was higher under sodium and fluorescent lighting compared to low- and high-pressure mercury lighting [9]. Recently, it was shown that the use of fiber optic lighting or a combination of fiber optic with standard fluorescent lighting significantly reduced greening progression more than either ceramic metal halide or halogen light sources [10]. Previous studies of light spectral quality have identified blue wavelengths to elicit a faster progression of greening compared to other wavelengths [4, 11, 12]. For example, higher greening propensity occurred when potato tubers were exposed to blue (475 nm) and red (675 nm) compared to green (525 nm) and yellow (575 nm) light [11]. This research was conducted using light filters [11, 12] or celluloid sheets of various colours [4]. The effect of specific light wavelengths on tuber greening requires further evaluation particularly using technologies such as light-emitting diodes (LEDs) that allow a more precise level of light manipulation.

Given that light is the major driver of potato greening, is it then possible to identify retail stores with high greening risk based on light quality and intensity alone? To explore this approach, this study firstly investigated the impact of specific light wavelengths on greening propensity in three potato varieties under laboratory conditions. Subsequently, measurement of the intensity and light quality of 25 retail stores was used to generate a greening risk rating for individual stores. A survey of the store managers from each retail store provided additional details on in-store management practices and factors that may also contribute to greening response, and thus potato stock loss.

## Materials and methods

### Plant material

Three potato varieties, Nicola, Maranca and Kennebec, known to have low, mid and high susceptibility to greening, respectively [13], were selected for the laboratory experiment. Seed tubers were propagated in 20 cm diameter plastic pots with potting mix containing sand, peat, and composted pine bark (10:10:80; pH 6.0) pre-mixed with Osmocote 16–3.5–10 NPK resin-coated fertilizer (Scotts Australia Pty Ltd. Baulkham Hills, Australia), and grown under controlled glasshouse conditions, between 18 and 24 ˚C. Soil was regularly topped up to protect growing tubers from light exposure. Each variety was harvested following natural senescence. Tubers that formed close to the soil surface were discarded while the remainder were stored in the dark at room temperature for approximately 30 days to allow post-harvest maturation. All selected tubers were of similar size and free of visible damage.

### Light exposure treatment

For each of the eight light treatments (full white light, 370, 420, 450, 530, 620, 660, and 735 nm; Table 1), three tubers of each variety were placed in separate growth chambers, equipped with spectral programmable LED lighting (RX30 Heliospectra, Sweden). Tubers were uniformly exposed for 120 consecutive hours, with a photosynthetic photon flux density of approximately 13 μmol m$^{-2}$ s$^{-1}$ at 20 ˚C.

### Colour assessment

Tuber colour was measured with a colorimeter (Konica Minolta CR-400, Osaka, Japan), standardized against a white tile, using L* (lightness), a* (green-red axis), and b* (blue-yellow axis)

**Table 1. Greening response of three potato varieties after 120 hours of light exposure to different light wavelengths.**

| Light Wavelengths (nm) | Varieties | ΔChlorophyll (mg L$^{-1}$) | Average Δ Chlorophyll (mg L$^{-1}$) | ΔE*ab | Average ΔE*ab |
|---|---|---|---|---|---|
| 450 (Blue) | Nicola | 3.23±0.38 | 4.73±0.66 a | 15.16±1.25 | 17.71±1.42 a |
| | Maranca | 3.70±0.19 | | 16.09±0.86 | |
| | Kennebec | 7.27±0.44 | | 21.89±1.27 | |
| (White) | Nicola | 1.37±0.04 | 2.89±0.49 b | 9.85±2.68 | 13.46±1.17 b |
| | Maranca | 2.61±0.15 | | 13.00±1.3 | |
| | Kennebec | 4.70±0.23 | | 17.55±1.5 | |
| 420 (Violet) | Nicola | 1.63±0.11 | 2.75±0.35 bc | 8.97±0.26 | 11.83±1.3 bc |
| | Maranca | 2.71±0.21 | | 10.06±0.44 | |
| | Kennebec | 3.92±0.34 | | 16.45±1.09 | |
| 530 (Green) | Nicola | 1.22±0.03 | 2.21±0.39 c | 5.98±0.97 | 8.78±1.18 cd |
| | Maranca | 1.67±0 | | 7.13±0.46 | |
| | Kennebec | 3.73±0.2 | | 13.22±0.59 | |
| 660 (Red) | Nicola | 1.05±0.18 | 1.65±0.22 d | 5.10±0.24 | 6.81±0.98 d |
| | Maranca | 1.61±0.23 | | 8.07±0.69 | |
| | Kennebec | 2.30±0.37 | | 7.25±1.41 | |
| 630 (Orange) | Nicola | 0.25±0.07 | 1.20±0.48 d | 4.16±0.42 | 7.22±1.63 d |
| | Maranca | 1.56±0.11 | | 7.79±1.57 | |
| | Kennebec | 1.78±0.12 | | 9.70±1.37 | |
| 370 (Ultraviolet) | Nicola | 0.40±0.12 | 1.14±0.20 d | 3.79±1.46 | 6.77±0.89 d |
| | Maranca | 1.58±0.13 | | 7.71±1.53 | |
| | Kennebec | 1.44±0.21 | | 8.82±2.12 | |
| 735 (Far red) | Nicola | 0.26±0.004 | 0.55±0.11 e | 2.81±0.51 | 4.17±0.44 e |
| | Maranca | 0.40±0.02 | | 4.27±0.61 | |
| | Kennebec | 0.68±0.29 | | 5.42±0.16 | |
| P value | | | <0.0001 | | <0.0001 |

Parameters assessed include changes in both chlorophyll content (ΔChlorophyll) and tuber skin colour (ΔE*ab). Data represent means ± SE. Different letters within each column indicate that means for each of ΔChlorophyll and ΔE*ab are significantly different between different light treatments at P <0.05 using Fisher's protected least significant difference (LSD) test.

parameters. Colour measurements were taken with three technical replicates, from the stem, the middle and the bud end of each tuber, immediately before and five days after light exposure treatment. Colour difference was calculated as follows:

$$\Delta E * ab = \left[ (\Delta L*)^2 + (\Delta a*)^2 + (\Delta b*)^2 \right]^{1/2}$$

Where $\Delta L^*$, $\Delta a^*$ and $\Delta b^*$ represent the differences in L*, a* and b* values before and after light treatments.

## Chlorophyll analysis

Three periderm disks from each tuber (1.5 mm thick and 1 cm diameter) were cut using a cork borer from the stem, the middle and the bud end of each tuber periderm. The disks were snap frozen in liquid nitrogen and ground to powder using a mortar and pestle. Samples were extracted with 5 mL of N, N-dimethylformamide for chlorophyll analysis. All samples were stored at 4 ˚C in the dark for 24 hours. After centrifugation for 15 min at 2500 × g, the absorbance was measured with a spectrophotometer (Thermo Scientific Spectronic 200E) at 647 and 664 nm. Chlorophyll concentrations were determined according to [14], using the below

equation and expressed in mg L$^{-1}$ fresh weight:

$$\text{Total chlorophyll} = 17.67(A647) + 7.12(A664).$$

## Assessment of lighting conditions in retail stores

A survey of 25 retail stores in Tasmania, Australia was conducted in October 2018 to assess the light intensity and spectra around the potato display area. The store survey was approved by the University of Tasmania Social Sciences Human Research Ethics Committee (Ref. H0017175). The lowest and highest light intensities (μmol m$^{-2}$ s$^{-1}$) were measured using a PAR light meter (Apogee MQ-500, Apogee Instruments, United States) positioned just above the displayed potatoes. Light quality was measured using a spectroradiometer (ASD HandHeld 2, Malvern Panalytical, Longmont CO USA), with 15 replications recorded for each store. Spectra were acquired in raw digital numbers (DN) for each wavelength.

## Potato losses in retail stores

Potato stock loss data for washed Nicola (2 kg bags), were obtained for each of the 25 retail stores for a period of 5 years (2014–2019). The stock loss reflected the percentage of bags that were considered not suitable for sale, predominantly due to tuber greening.

## Survey of store managers

Of the 25 retail stores, 17 store/produce managers completed the online surveys. The survey consisted of ten questions (S3 Fig) aimed at identifying managers' perception of greening and management strategies employed in stores to reduce greening.

## Statistical analysis

For the light exposure in growth chamber experiments, the effect of light wavelengths on greening (Δ chlorophyll and ΔE*ab) of different varieties were analysed using two-way analysis of variance. Means were separated using Fisher's protected least significant difference (LSD) test ($p < 0.05$). Analyses were conducted using R version 3.6.1 [15].

For the store lighting assessment experiments, spectra from all stores were standardised by subtracting the minimum value per replicate from each spectrum, and then averaging the replicated spectra for each store. In order to generate a risk profile for each store, based on lighting conditions, we firstly generated weighting functions for Nicola, Maranca and Kennebec (S1 Fig) using the chlorophyll concentrations at specific light wavelengths (370, 420, 450, 530, 620, 660, and 735 nm), as determined from the controlled experiment. Chlorophyll values were linearly interpolated for all wavelengths (325–735 nm) between each pair of points (S1 Fig). Secondly, the weighting function was multiplied by the average spectra obtained in each store, resulting in a 'weighted luminance area'. The area under the 'weighted luminance' profile was considered as the greening risk metric for each store.

A linear regression approach was used to assess the relationships between greening risk metric and i) light intensity; ii) ceiling height; and iii) historical potato stock loss in stores for Nicola. Software used was PROC REG in SAS version 9.4.

# Results

## Greening response to different light wavelengths

Significant differences were found in greening response between tubers exposed to different light wavelengths, with comparable responses observed for greening parameters measured,

chlorophyll concentration and tuber skin colour. For all three cultivars, maximum greening occurred when tubers were exposed to 450 nm (blue), followed by white, 420 (violet) and 530 nm (green), and then 660 nm (red), 620 (orange) and 370 (ultraviolet), while the lowest greening rates were obtained at 735 nm (far red) (Table 1).

### Light intensity in retail stores

Light intensity, just above the potato surface on store shelves, varied between the 25 retail stores, with one store (store T) having a range of intensities as low as 3–6 µmol m$^{-2}$ s$^{-1}$, while another (store K) having intensities as high as 19–46 µmol m$^{-2}$ s$^{-1}$ across readings (Fig 1). The average intensity varied from 4.5 to 32.5 µmol m$^{-2}$ s$^{-1}$.

### Light spectra in retail stores

As revealed by the spectral data, most stores were dominated by two primary emission peaks in the blue and orange regions, at approximately 450 and 600 nm, respectively, while a few stores showed three major emission peaks in the blue, green and orange regions, at approximately 450, 545 and 610 nm, respectively (Fig 2; S1 File). Two stores showed an additional peak in the far-red region at around 820 nm. There was substantial variability in the intensity of emissions between stores (S1 File).

### Greening risk rating based on lighting conditions in retail stores

Based on quantitative analysis of lighting conditions, substantial differences in greening risks were found across stores for each of the three varieties, with some (e.g. Stores O, B, P and E) having around three-fold higher risk than others (e.g. Store T) (Fig 3). The highest greening risk was recorded for Kennebec, followed by Maranca then Nicola (Fig 3).

In general, greening risk was found to be related to light intensity level in stores (R$^2$ = 0.48; p<0.001; Fig 4). Ceiling heights (S1 Table) showed no relationship with greening risk (S2 Fig). Similarly, there appeared to be no relationship between light spectral pattern and greening risk.

### Survey with store managers

Of the 17 out of 25 online surveys completed, greening was identified as an issue by 65% of store managers resulting in tuber waste. Further, 16 out of the 17 store managers believed that

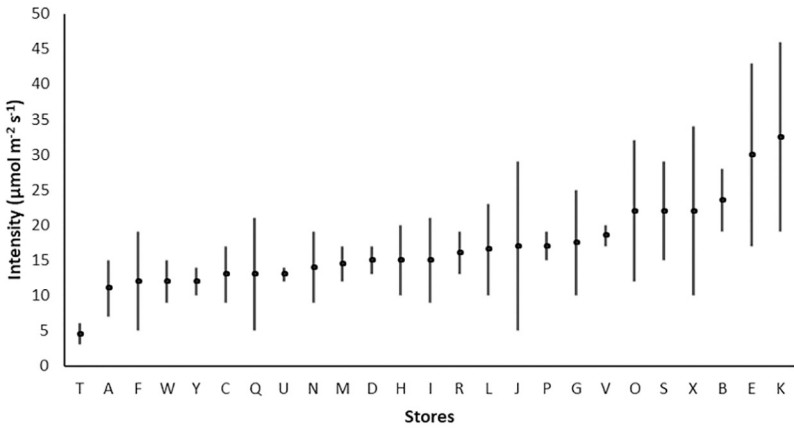

**Fig 1. Average light intensity in 25 retails stores (A-Y) measured at potato display height.** The vertical bar shows the lowest and highest light intensities at each store.

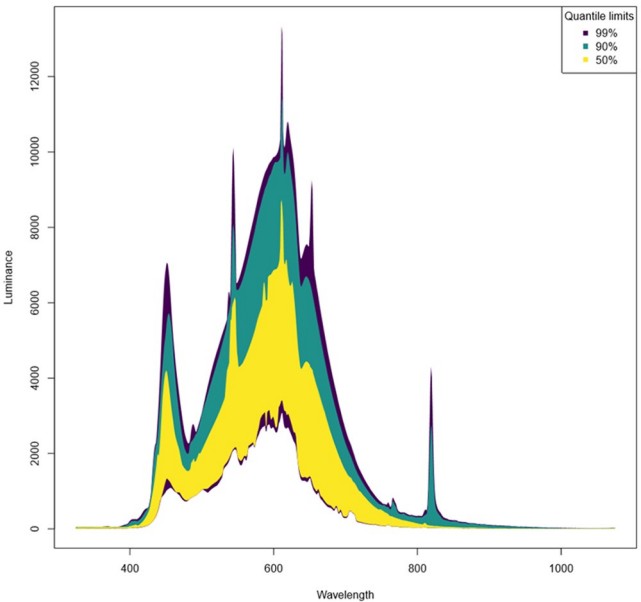

**Fig 2. Envelopes for standardized light spectra in 25 retail stores (A-Y) showing the quantile limits containing the inner 99%, 90% and 50% of the data at each wavelength.** Spectral data were obtained by subtracting the minimum value per replicate from each spectrum, and then calculating the upper quantile limit for each store.

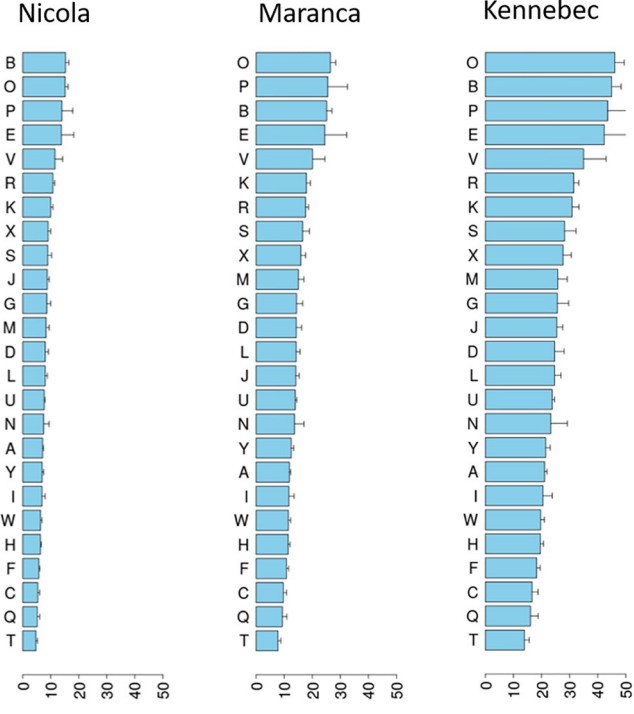

**Fig 3. Greening risk rating based on lighting conditions in 25 retail stores (A-Y) for three potato varieties.** Risk rating incorporates light spectral patterns (an average of 15 spectral measurements taken randomly at potato shelf height) coupled with greening rates, at specific light wavelengths, obtained under controlled experimental conditions.

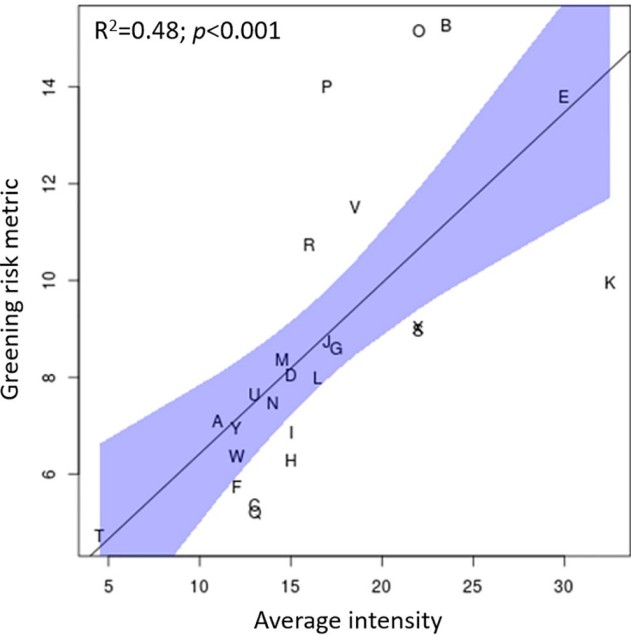

**Fig 4. Fitted regressions with observed data of average intensity versus greening risk.** The shaded areas are the 95% confidence and prediction intervals. Each letter represents an individual retail store.

greening was the main reason for customer rejection of potatoes (S3 Fig). Stores had varying lighting and display practices with the number of hours of illumination ranging from 14 to 24 hours per day (S3 Fig). Although all stores covered potatoes displayed on store shelves overnight, the material used differed between stores, including tarpaulin, plastic covers, cloth and bin liners, while in the storage area, 71% of stores were found to cover potato bins (S3 Fig). Prior to being displayed, potatoes could be stored for up to two days and once on display, they could remain on the shelf for up to two days. The majority (71%) of store managers noticed that the highest greening rates occurred during summer (S3 Fig).

## Potato losses in stores

The percentage of discarded Nicola 2 kg bags varied across stores over the 5-year data period, with losses ranging from 0.8% to 5.6% (Table 2). The average loss across all stores was 2.3%. Despite a low $R^2$ of 0.2, there was a significant positive linear relationship between greening risk and potato stock loss (Stockloss = 2.9+0.35*Greening risk; p<0.05) (Fig 5).

## Discussion

Under controlled conditions, we showed tuber greening, for all three potato varieties, was greatest under blue light with 53%, 65% and 75% less greening occurring under green, red and orange light, respectively. We coupled these findings with direct in-store measurements of lighting conditions to develop a tuber greening risk rating model. Significant differences in greening risk based on measured lighting were found across stores. The tuber greening risk metric was able to partially explain historic average potato stock loss in stores.

Our finding that the highest chlorophyll concentrations were found at 450 nm (blue), is in accordance with the results of [11]. However, they also reported that 675 nm (red) was effective for synthesis of chlorophyll a and b [11]–in contrast, we found greening was three-fold

**Table 2. Percentage of stock loss for Nicola (2 kg bags) for 25 retail stores (A-Y).**

| Stores | Stock loss (%) |
|--------|----------------|
| A | 1.8 |
| B | 5.6 |
| C | 1.7 |
| D | 1.5 |
| E | 1.9 |
| F | 2.2 |
| G | 4.0 |
| H | 2.7 |
| I | 1.3 |
| J | 0.8 |
| K | 0.9 |
| L | 1.9 |
| M | 4.7 |
| N | 1.9 |
| O | 3.2 |
| P | 2.6 |
| Q | 1.7 |
| R | 3.4 |
| S | 3.4 |
| T | 1.0 |
| U | 1.7 |
| V | 1.7 |
| W | 1.6 |
| X | 2.5 |
| Y | 2.9 |
| Average | 2.3 |

Data represent the average stock loss for 5 years (2014–2019).

greater at 450 nm than at 660 nm (red). This may be attributed to the broad-band light filters used in earlier studies, which could transmit a mixture of wavelengths that may have confounded results. The intense greening at blue wavelengths observed in this study is consistent with previous studies in leaf tissues, which showed enhanced activity of key chlorophyll enzymes [16, 17], leading to higher chlorophyll accumulation as compared to red light [18–21].

Light intensity varied between different stores and between different locations around the potato display area in each store, ranging from 3 to 46 µmol m$^{-2}$ s$^{-1}$. In a survey of seven retail outlets in Washington USA, potatoes were found to be displayed at lower light intensities, ranging from 2 to 10 µmol m$^{-2}$ s$^{-1}$ [22]. Further, most of the stores also had tubers in lighted displays on refrigerated shelves, with an average light intensity of 28 µmol m$^{-2}$ s$^{-1}$ [22]. These cooler temperatures could assist in reducing greening propensity [2], however, these temperatures may cause cold-induced sweetening [23], which could be a concern for fried potato products. In this study, all potatoes were displayed at room temperature (around 20 ˚C) during our store assessment.

The novel tuber greening risk rating indicated some of the surveyed retail stores had up to a three-fold greater risk of greening than others. The increase in greening risk was mainly related to an increase in light intensity, but not to light quality or the height of lighting systems

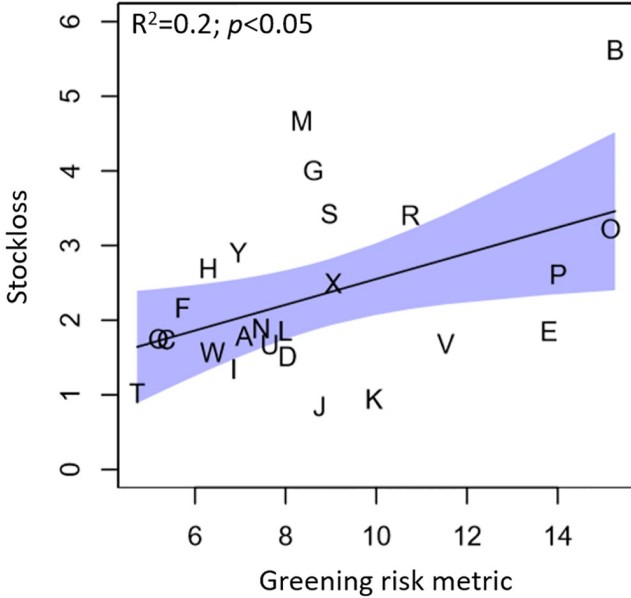

**Fig 5. Fitted regressions with observed data of stock loss of Nicola versus greening risk metric.** The shaded areas are the 95% confidence and prediction intervals. Each letter represents an individual retail store.

above potato displays. The relationship between tuber greening risk rating and historic stock losses while significant, had a low $R^2$ value indicating other in-store factors, such as those identified in the online survey are of importance. For example, despite a very high average light intensity of 32.5 $\mu$mol m$^{-2}$ s$^{-1}$, the low stock loss of stores K was surprisingly low (0.9%) (Table 2), and in this case, may be due to the reduced illumination hours and fast sale turnover (S3 Fig) rather than light intensity. This finding suggests that for stores identified as vulnerable to greening risk due to lighting conditions can use a range of store management options to mitigate this greening risk.

Management practices and day-to-day operational decisions in stores can have a major impact on greening propensity, and potato stock loss primarily related to the risk of light exposure to tubers. These may include variety selection, stock control and turnover, shop opening hours and methods of displaying produce. Our online survey of store managers provided a snapshot of how some of these factors were managed and identified and the variation between Tasmanian stores. For example, potatoes in stores were exposed to light from half a day up to two days. In contrast, longer exposure times, ranging from 1 to 2 days to 1–2 weeks, were reported in a telephone survey of six produce managers at major chain groceries in Wisconsin USA [1]. Likewise, variations in the duration of lighting were found between different retail stores, which ranged from 14 to 24 hours per day. Furthermore, whether potatoes are displayed in bags or loose could affect the level of light reaching tubers and therefore, is another factor affecting in-store greening. Our survey also reported that greening was higher in the summer months with longer daylengths, during which potatoes in retail stores are either from the end of season stock, with long storage duration, or from the early harvested potato stock, with less mature skin and as such higher greening tendency [13].

## Conclusions

In conclusion, our study generated a novel potato tuber greening risk rating of retail stores based on lighting conditions, which impact potato stock losses, and can be influenced by store

management practices. Future models, predicting in-store greening, could be expanded to take into consideration other store conditions and management practices that may affect greening such as temperature, lighting duration, average time till sale, and light protection during non-retail periods. As perceived by store managers, our survey showed that greening was identified as the primary reason for customers not purchasing potatoes, in accordance with a previous survey in the USA [1]. In order to reduce in-store greening, future research should consider the manipulation of lighting conditions, by testing different combinations of light wavelengths to determine the optimal spectral composition for reducing greening propensity. The LED technology [24] could potentially be designed to avoid specific wavelengths, such as blue light, to limit greening. Furthermore, innovative packaging types, that limit the level of photosynthetically active radiation reaching potatoes and/or filter blue wavelengths may offer potential to control or reduce greening on store shelves while enhancing their appeal to the consumer.

## Supporting information

**S1 Fig. Weighting factor used to calculate greening risk.** The weighting factor was calculated using the chlorophyll concentration of Nicola, after 120 hours of light exposure, at different light wavelengths (370, 420, 450, 530, 620, 660, and 735 nm). Given the observed chlorophyll, values were linearly interpolated for all wavelengths (325–735 nm) between each pair of points. (TIF)

**S2 Fig. Fitted regression with greening risk metric versus ceiling height.** The shaded areas are the 95% confidence and prediction intervals. Each letter represents an individual retail store. (TIF)

**S3 Fig. Survey questionnaire on potato greening with 17 retail store/produce managers in Tasmania.** Each letter represents an individual retail store. (TIF)

**S1 Table. Distance between the stores ceilings and the potato shelf displays in 25 Tasmanian retail stores.** Data represent the closest and furthest distance measured. Values are presented in meters (m). (TIF)

**S1 File. Unstandardized light spectra in 25 Tasmanian retail stores.** In each store, around 15 measurements were taken randomly around the potato display area, at shelf height, shown as multiple spectra in each graph. Raw DN refers to raw digital numbers. (PDF)

## Acknowledgments

We thank Dr Jim Weller for access to and support with the LED lighting in the growth chambers and Dr Juliane Bendig for technical assistance with the spectroradiometer.

## Author Contributions

**Conceptualization:** Sabine Tanios, Alieta Eyles, Robert S. Tegg, Calum R. Wilson.

**Data curation:** Sabine Tanios, Ross Corkrey, Calum R. Wilson.

**Formal analysis:** Sabine Tanios, Ross Corkrey.

**Funding acquisition:** Calum R. Wilson.

**Investigation:** Sabine Tanios, Alieta Eyles, Robert S. Tegg, Tamilarasan Thangavel.

**Methodology:** Sabine Tanios, Alieta Eyles, Ross Corkrey, Robert S. Tegg, Tamilarasan Thangavel, Calum R. Wilson.

**Project administration:** Calum R. Wilson.

**Resources:** Calum R. Wilson.

**Supervision:** Alieta Eyles, Robert S. Tegg, Tamilarasan Thangavel, Calum R. Wilson.

**Validation:** Alieta Eyles, Ross Corkrey, Robert S. Tegg, Calum R. Wilson.

**Writing – original draft:** Sabine Tanios.

**Writing – review & editing:** Sabine Tanios, Alieta Eyles, Ross Corkrey, Robert S. Tegg, Tamilarasan Thangavel, Calum R. Wilson.

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
