## [Decision Letter · Decision Letter 0]

1 Jul 2020

PONE-D-20-17658

Quantifying risk factors associated with light-induced potato tuber greening in retail stores

PLOS ONE

Dear Dr. Wilson,

Thank you for submitting your manuscript to PLOS ONE. After careful consideration, we feel that it has merit but does not fully meet PLOS ONE’s publication criteria as it currently stands. Therefore, we invite you to submit a revised version of the manuscript that addresses the points raised during the review process.

We look forward to receiving your revised manuscript.

Kind regards,

Vijay Kumar

Academic Editor

PLOS ONE

Journal Requirements:

2. Please ensure that you refer to Figure 5 in your text as, if accepted, production will need this reference to link the reader to the figure.

Additional Editor Comments (if provided):

Reviewers' comments:

Reviewer's Responses to Questions

**Comments to the Author**

1. Is the manuscript technically sound, and do the data support the conclusions?

Reviewer #1: Yes

2. Has the statistical analysis been performed appropriately and rigorously? 

Reviewer #1: Yes

3. Have the authors made all data underlying the findings in their manuscript fully available?

Reviewer #1: Yes

4. Is the manuscript presented in an intelligible fashion and written in standard English?

Reviewer #1: Yes

5. Review Comments to the Author

Reviewer #1: The manuscript is well written; However I have a few point to raise: 1) Is light condition the only issue in potato greening? If not, the other factors should be taken into considerations and the paper would have been more exciting if the cumulative effects of all the factors were analyzed. Besides, while measuring the greening of potato under storage conditions, were the other parameters were uniform to obtain uniform results? 2) The methodology section describes the varied chlorophyll content in such potato tubers; However, the other nutritional parameters are needed to be addressed in order to get a wide picture on the effects of light on greening and nutritional quality of the tubers. There are simple analyses by which the starch content, mineral content, antioxidative ability etc. can be measured across potato collections.

6. PLOS authors have the option to publish the peer review history of their article (what does this mean?). If published, this will include your full peer review and any attached files.

Reviewer #1: **Yes: **Abhijit Dey

---

## [Author Response · Author response to Decision Letter 0]

9 Jul 2020

Response to Editor and Reviewers Comments

Response: 

Yes – our manuscript and file names meet the formatting guidelines. 

2. Please ensure that you refer to Figure 5 in your text as, if accepted, production will need this reference to link the reader to the figure.

Response: 

We have included a reference to Fig 5 in the text (which was erroneously previously cited as 4B)

Comments to the Author

1. Is the manuscript technically sound, and do the data support the conclusions?

Reviewer #1: Yes

Response: no response required

2. Has the statistical analysis been performed appropriately and rigorously? 

Reviewer #1: Yes

Response: no response required

3. Have the authors made all data underlying the findings in their manuscript fully available?

Reviewer #1: Yes

Response: no response required

4. Is the manuscript presented in an intelligible fashion and written in standard English?

Reviewer #1: Yes

Response: no response required

5. Review Comments to the Author

Reviewer #1: The manuscript is well written; However I have a few point to raise: 

1) Is light condition the only issue in potato greening? If not, the other factors should be taken into considerations and the paper would have been more exciting if the cumulative effects of all the factors were analyzed. 

Response: 

Light (intensity and specific light wavelength as demonstrated here) is the major environmental factor driving potato tuber greening. Management factors that can influence tuber greening considered in the manuscript (stock turnover, packaging, bin coverings etc) are again related to levels of light exposure. 

There are potential very minor effects of temperature and oxygen levels (Tanios et al 2018), but (a) in this study the tuber greening assessments were all conducted under controlled conditions with uniform temperature and atmospheric conditions, and (b) in a retail environment these parameters do not vary considerably. There does exist significant differences between potato varieties in their propensity of tuber greening as is shown in this study and in many prior studies. From our previous work varietal differences are associated with the level of suberin deposition within tubers (Tanios et al 2020).

We thus developed a risk rating model for retail stores based on the lighting conditions employed in potato display areas. The model was generated for three different varieties, known to have low, mid and high susceptibility to greening. 

We identify retail store management practices that are likely to impact tuber light exposure but did not include these within the light exposure risk model developed as these variables are, in general, uncontrolled and often impossible to measure. The major point of the manuscript was to show that retail store lighting systems provide a set risk based on light intensity and wavelength that is then attenuated or exacerbated by good or poor management practices. This is further discussed in the revised manuscript (lines 280-283). We note that attempting to consider management factors in the risk model we present is not possible and is clearly beyond the scope of the current work.

Besides, while measuring the greening of potato under storage conditions, were the other parameters were uniform to obtain uniform results?

Response: 

Greening was not measured for the model development under storage conditions nor in retail stores. It was measured under a strictly controlled environment and uniform conditions following the exposure to seven different light wavelengths. Tubers were incubated in growth chambers, equipped with spectral programmable LED lighting. Apart for differing light wavelength treatments, fully uniform conditions were maintained in growth chambers, including duration and intensity of lighting provide and temperature. This has been now further clarified in the revised manuscript.

2) The methodology section describes the varied chlorophyll content in such potato tubers; However, the other nutritional parameters are needed to be addressed in order to get a wide picture on the effects of light on greening and nutritional quality of the tubers. There are simple analyses by which the starch content, mineral content, antioxidative ability etc. can be measured across potato collections.

Response: 

The manuscript has the clear focus of assessing the risk of lighting systems on potato tuber greening, which is regarded as the major source of stock loss in the retail sector for fresh potatoes. Assessing the effect of light on other nutritional parameters may have some interest from a nutritional perspective but is clearly beyond the scope of the present study and would not greatly add to major findings of the manuscript, which quantified risk factors associated with potato tuber greening in retail stores and developed a greening risk rating model for retail stores based on lighting conditions employed in potato display areas. 

6. PLOS authors have the option to publish the peer review history of their article (what does this mean?). If published, this will include your full peer review and any attached files.

Do you want your identity to be public for this peer review? For information about this choice, including consent withdrawal, please see our Privacy Policy.

Reviewer #1: Yes: Abhijit Dey

Response: no response required

Final Note: 

We have uploaded our figures to the Preflight Analysis and Conversion Engine (PACE) digital diagnostic tool as required and have ensured each pass PACE standards. We did not run the supplementary PDF file through PACE as it converts the file into 5 separate images which is against journal requirements to have these as a single file

References cited

Tanios S, Eyles A, Tegg R, Wilson C. Potato tuber greening: a review of predisposing factors, management and future challenges. Am. J. Potato Res. 2018;95: 248-57.

Tanios S, Thangavel T, Eyles A, Tegg R, Nichols D, Corkrey R, Wilson C. Suberin deposition in potato periderm: a novel resistance mechanism against tuber greening. New Phytol. 2020;225: 1273-1284.

---

## [Editor Report · Decision Letter 1]

15 Jul 2020

Quantifying risk factors associated with light-induced potato tuber greening in retail stores

PONE-D-20-17658R1

Dear Dr. Wilson,

We’re pleased to inform you that your manuscript has been judged scientifically suitable for publication and will be formally accepted for publication once it meets all outstanding technical requirements.

Kind regards,

Vijay Kumar

Academic Editor

PLOS ONE
---

## [Editor Report · Acceptance letter]

17 Jul 2020

PONE-D-20-17658R1 

Quantifying risk factors associated with light-induced potato tuber greening in retail stores 

Dear Dr. Wilson:

I'm pleased to inform you that your manuscript has been deemed suitable for publication in PLOS ONE. Congratulations! Your manuscript is now with our production department. 

Kind regards, 

on behalf of

Dr. Vijay Kumar 

Academic Editor

PLOS ONE